# Motivational Factors in Intergenerational Sustainability Dilemma: A Post-Interview Analysis

**Mostafa E. Shahen** [1,2,3]**, Wada Masaya** [4]**, Koji Kotani** [1,2,5,6]*⬛ **and Tatsuyoshi Saijo** [1,2,7]⬛

[1]  School of Economics and Management, Kochi University of Technology, 2-22 Eikokuji-cho, Kochi-shi,
    Kochi 780-8515, Japan; 226002a@gs.kochi-tech.ac.jp (M.E.S.); tatsuyoshisaijo@gmail.com (T.S.)

[2]  Research Institute for Future Design, Kochi University of Technology, A401 2-7-8 Otesuji, Kochi-shi,
    Kochi 780-0842, Japan

[3]  Faculty of Commerce, Zagazig University, Shaibet an Nakareyah, Zagazig 44519,
    Ash Sharqia Governorate, Egypt

[4]  Ginken LTD, Tokyo 135-0063, Japan; wada.m@ginken.com

[5]  Urban Institute, Kyusyu University, Fukuoka 819-0395, Japan

[6]  College of Business, Rikkyo University, Tokyo 171-8501, Japan

[7]  Research Institute for Humanity and Nature, Kyoto 603-8047, Japan

*  Correspondence: kojikotani757@gmail.com

**Abstract:** An intergenerational sustainability dilemma (ISD) is a situation of whether or not a person sacrifices herself for future sustainability. However, little is known about what people consider while making a decision under ISD. This paper analyzes motivational factors for people to decide under ISD, hypothesizing that the factors can be different with or without perspective-taking of future generations. One-person basic ISD game (ISDG) along with post-interviews are instituted where a lineup of individuals is organized as a generational sequence. Each individual chooses an unsustainable (or sustainable) option with (without) irreversibly costing future generations in 36 situations. A future ahead and back (FAB) mechanism is applied as a treatment for perspective-taking of future generations where each individual is asked to take the next generation's position and to make a request about the choice that he/she wants the current generation to choose, and next, he/she makes the actual decision from the original position. By analyzing the post-interview contents with text-mining techniques, the paper finds that individuals mostly consider how previous generations had behaved in basic ISDG as the main motivational factor. However, individuals in FAB treatment are induced to put more weight on the possible consequences of their decisions for future generations as motivational factors. The findings suggest that perspective-taking of future generations through FAB mechanism enables people to change not only their behaviors but also motivational factors, enhancing ISD.

**Keywords:** content analysis; future ahead and back mechanism; future design; intergenerational sustainability dilemma; text-mining

---

## 1. Introduction

The intergenerational sustainability dilemma (hereafter, ISD) is a situation where individuals choose to maximize (or sacrifice) their benefits without (for) considering future generations, compromising (maintaining) intergenerational sustainability (hereafter, IS) [1,2]. Democracy and capitalism are two main institutions that have been adopted by many countries around the world, while the decision-making process under these institutions does not incorporate the perspective of future generations [3–5]. This absence of future generations in the current political system and the unidirectional nature of ISD lead to exploitation of resources by the current generation

without considering future ones, causing several IS problems such as the accumulation of public debt, climate change and depletion of natural resources [6–11]. To solve these IS problems, some mechanism or institution should be applied to enable the current generation to take the perspective of future generations.

Some researchers advocate a new field of research to addresses such IS problems, which is called "future design", suggesting several mechanisms such as future ahead and back (FAB), intergenerational accountability and imaginary future generations to connect the current and future generations [1,2,12,13]. Several studies have analyzed people's behaviors in ISD by applying economic experiments, finding that the introduction of future design mechanisms induce people to enhance IS [1,2,14]. These individual and group behavioral changes can be a result of a shift in the motivations (i.e., the decision making context, scrutiny of others and cultural backgrounds) by the introduction of future design mechanisms [15]. Thus, this paper aims to explore the motivational factors of individual behaviors in ISD.

Several researchers examine the group behavior towards IS in laboratory experiments. Fischer et al. [16] employ a common pool resource experiment to investigate individual decisions in a group using a subject pool of students, finding that the subsequent group's existence enhances individual sustainable behaviors. Hauser et al. [17] institute an online intergenerational goods experiment with a voting mechanism by recruiting general people as subjects and demonstrate that the introduction of voting allows cooperators to restrain the defectors from resource exploitation. Sherstyuk et al. [18] examine the efficiency of dynamic externalities, showing that resolving the dynamic externalities is less challenging in infinitely lived decision-maker settings than in intergenerational ones. Fochmann et al. [19] study the role of intergenerational altruism to solve IS problem i.e., accumulation of public debt, indicating that fairness concern for the future generation can not maintain IS. These studies find that, in general, IS problems are complicated and difficult to resolve with a simple intervention such as intergenerational altruism.

Other scholars have focused on the group behavior in ISD in laboratory and field experiments. Kamijo et al. [1] design and implement a laboratory experiment of ISD game (ISDG) using a student pool to understand the group behaviors in ISD, claiming that the introduction of a negotiator for future generations (i.e., imaginary future person) enhances IS. Shahrier et al. [2,20] use a subject pool of general people to conduct an ISDG field experiment in Bangladesh, showing that rural people maintain IS more often than urban ones do due to a high proportion of prosocial people. They also introduce an institution called "future ahead and back (FAB) mechanism" that induces subjects to take the perspective of the next generation before making their decisions, which improves IS. These papers show laboratory experiments can reveal the factors that affect group behaviors in ISD such as prosociality and a new institution i.e., FAB. These experimental methods of research are not enough to explore the underlying motivational factors that might affect the group as well as individual behaviors.

Thus, few studies implement experimental methods along with qualitative methods of post-interviews to study how groups and individuals make decisions in various settings [21,22]. Timilsina et al. [23] conduct an ISDG field experiment and a post-interview with a subject pool of general people in Nepal to examine the effect of deliberation on people's opinions. They find that urban people's opinions do not change after deliberation to maintain IS compared to rural people. The post-interview analysis is implemented by Castillo et al. [24] in a fishery field experiment to understand the effect of life experience on the decision made by rural communities in Colombia and Thai. They demonstrate that rural people perceive the local government as an obstacle to the proper governance of fishery resources. Butler et al. [25] apply a post-interview analysis with prisoner's dilemma, chicken, dictator and ultimatum games in laboratory experiments with a student subject pool to study the perception of the tension between cooperative and selfish impulses. They show that selfish motives, emotions and safety concerns are the main motivational factors that drive individual

behaviors. Overall, these studies indicate that the opinions, perceptions and motivational factors of individual behaviors in experiments can be explored by analyzing post-interviews.

Most of the literature has focused on the group behaviors in ISD, while few studies focus on the way to elicit individual opinions, perceptions and motivational factors in some economic games using a post-interview analysis. However, none of the previous studies have analyzed a post-interview to investigate the motivational factors in ISD. Given this state of affairs, this paper seeks to identify the motivational factors for people to make their decision under ISD, hypothesizing that the factors can be different with or without perspective-taking of future generations. Thus, this paper institutes a one-person ISD game (ISDG) where a lineup of individuals is organized as a generational sequence. Each individual chooses an unsustainable (or sustainable) option with (without) irreversibly costing future generations in 36 situations where previous generations' choices and payoff structures are parameterized. A FAB mechanism is applied as a treatment to resolve ISDG through perspective-taking of future generations where each individual is asked to take the next generation's position and to make a request about the choice that he/she wants the current generation to choose, and next, he/she makes the actual decision from the original position. After the game, each individual is interviewed for 7~10 min, enabling us to analyze the post-interview contents. The novelty of this research lies in implementing a content analysis for the post-interview by text-mining techniques to reveal the motivational factors for individuals with and without taking the perspective of future generations in ISD. This paper is organized as follows. Section 2 lays out the methods and materials applied in this research. Section 3 explains the analytical results. Section 4 discusses the reasons and implications for such results. Section 5 concludes the paper and provides the limitations.

## 2. Methods and Materials

### 2.1. Experimental Setup

The experiment is administered at the Kochi University of Technology (KUT) experimental laboratories. This experiment consists of 27 sessions, each involving 4~5 subjects with a total of 104 subjects. The subject pool of this study is undergraduate students of KUT, while the sample represents around $(104/2304) \times 100 = 4.5\%$ of the subject pool. This sample has the same distribution of the subject pool of KUT students because the numbers of males and females in this sample are 49 and 55, respectively. The average age in the sample is 20.4. A one-person "intergenerational sustainability dilemma game" (ISDG) and a post-interview are conducted to collect the data of subjects' decisions and their motivational factors in ISDG.

One-Person Intergenerational Sustainability Dilemma Game (One-Person ISDG)

A one-person ISDG is designed and instituted with a similar structure of ISDG played by a group of three people in Kamijo et al. [1] and Shahrier et al. [2]. In one-person ISDG, a lineup of individuals is organized as a generational sequence where each generation is represented by one person. Each generation is requested to make a choice between an unsustainable option $A$ and a sustainable option $B$. When a generation chooses option $A$, he/she receives $X$ tokens (hereafter, "tokens" is not mentioned) as a payoff and the next generation's payoffs associated with options $A$ and $B$ uniformly decrease by $D$. When a generation chooses option $B$, he/she receives $X - D$ as a payoff and the next generation's payoffs associated with options $A$ and $B$ remain the same as the current one. This represents an essential feature of ISDG because the current generation affects the welfare of subsequent generations, while the opposite is not true. In one-person ISDG, the 1st generation always starts the game with options $A = 3600$ and $B = 3600 - D$ in all situations. When a subject is the 1st generation and plays the game with $D = 900$ in one situation, he/she receives 3600 by choosing option $A$ and the 2nd generation plays the game with options $A = 2700$ and $B = 1800$. He/she receives 2700 by choosing option $B$, and the 2nd generation plays the game with options $A = 3600$ and $B = 2700$.

A strategy method is applied to create 36 different situations of one-person ISDG that each subject goes through [26]. The strategy method applied in this research follows Bardsley [27], which they call a conditional information lottery (CIL) method. This method enables us to create several fictional situations and one real situation, where subjects can not distinguish between the real and fictional ones. The 36 situations consist of 35 fictional situations, which are the same for all the subjects, and one real situation (i.e., binding situation) that varies across subjects. In the 35 situations, the history of previous generations' choices and their number (i.e., history), the payoff of $X$ a generation can receive, a payoff difference of $D$ between options $A$ and $B$ and the ratio between $X$ and $D$ (i.e., $\frac{X}{D}$) are parameterized under the assumptions that the 1st generation always starts one-person ISDG with options $A = 3600$ and $B = 3600 - D$ as well as the value of $D$ remains the same in each situation.

Table 1 summarizes the 35 different situations of one-person ISDG, listing the associated percentages of previous generations that choose unsustainable option $A$ in history ranging from 0 to 1, the number of previous generations ranging from 0 to 23, the payoff a generation can receive $X$ ranging from 0 to 3600 and the difference $D$ ranging from 100 to 1800 and the ratio between $X$ and $D$ raging from 0 to 36. $\frac{X}{D}$ has an important meaning in ISDG because it represents how many generations can enjoy the positive amount of resources before reaching the resource extinction (i.e., $X = 0$), when all the current and subsequent generations keep choosing unsustainable options. Although Table 1 contains the percentage of previous generations in history for each situation that chose option $A$ as a summary, a subject is shown a whole history of how each previous generation chose options $A$ or $B$ displayed by a sequence of human shaped icons with different colors in each situation, respectively.

In addition to these 35 situations of one-person ISDG, each subject plays one binding situation whose decision environments evolve over generations according to how previous generations have chosen and how the current generation chooses, being actually passed to the subsequent generations within a sequence for the real payment to subjects. In the binding situation, the 1st generation starts the game with option $A = 3600$, where one value of $D$ is randomly picked out of the four possible values of 300, 600, 900 and 1200. Once it is picked, the value of $D$ remains the same for the 1st, 2nd, ... generations in a sequence for the binding situation. This situation is continued as far as the value of $X$ is strictly positive, and ends when it becomes zero or negative at some generation in a sequence. Therefore, the payoff structures in the decision environment each generation has faced in a sequence for the binding situation are different, while 35 situations in Table 1 are uniformly played by all subjects. A series of experimental procedures that asks each subject to play one-person ISDG in 36 situations is called "basic ISDG" treatment.

Building upon the basic ISDG treatment, the future ahead and back (FAB) treatment is prepared by applying the FAB mechanism for one-person ISDG in 36 situations. In FAB treatment, each subject is asked to go through the following steps in each situation. As the 1st step, each subject is asked to imagine that he/she is in the next generation. As if he/she is in the next generation, he/she is asked to make a request about the choice that he/she wants the previous generation to choose between options $A$ and $B$. As the 2nd step, the subject is asked to return back to her original (actual) position in the sequence and he/she makes her final and actual decision by choosing one option, $A$ or $B$ for that situation. For instance, if a subject is the 5th generation in a sequence for one situation, then he/she is asked to imagine herself in the position of the 6th generation in the sequence, and he/she is asked to make a request about the choice that he/she wants the 5th generation to take in the sequence. After that, he/she is asked to go back to her original position (i.e., the 5th generation) in the sequence and makes her final and actual choice for that situation. Each subject is randomly assigned to either basic ISDG treatment or FAB treatment and plays one-person ISDG with a strategy method in 36 different situations consisting of 35 situations in Table 1 and a single binding situation. The orders of 36 situations each subject goes through in one-person ISDG are randomly shuffled to avoid the order effects. In one-person ISDG, one experimental token is calculated and exchanged to be 1.5 JPY, and subjects are paid 3000 JPY ($\approx$ 27.8 USD) on an average.

**Table 1.** The 35 situations that each subject plays in one-person intergenerational sustainability dilemma game (ISDG).

| Situations | % of Option A in History | # of Generations in History | X | D | $\frac{X}{D}$ |
|:---:|:---:|:---:|:---:|:---:|:---:|
| | History | | | | |
| 1 | 0 | 0 | 3600 | 1800 | 2 |
| 2 | 0 | 5 | 3600 | 1200 | 3 |
| 3 | 0 | 7 | 3600 | 900 | 4 |
| 4 | 0 | 0 | 3600 | 300 | 12 |
| 5 | 0 | 9 | 3600 | 100 | 36 |
| 6 | 0.25 | 4 | 2700 | 900 | 3 |
| 7 | 0.25 | 8 | 1800 | 300 | 6 |
| 8 | 0.25 | 4 | 3400 | 200 | 17 |
| 9 | 0.33 | 9 | 0 | 1200 | 0 |
| 10 | 0.33 | 12 | 1200 | 600 | 2 |
| 11 | 0.5 | 4 | 0 | 1800 | 0 |
| 12 | 0.5 | 8 | 0 | 900 | 0 |
| 13 | 0.5 | 4 | 1200 | 1200 | 1 |
| 14 | 0.5 | 4 | 2400 | 600 | 4 |
| 15 | 0.5 | 4 | 2400 | 600 | 4 |
| 16 | 0.5 | 8 | 2400 | 300 | 8 |
| 17 | 0.5 | 2 | 3400 | 200 | 17 |
| 18 | 0.5 | 8 | 3200 | 100 | 32 |
| 19 | 0.63 | 8 | 2600 | 200 | 13 |
| 20 | 0.67 | 3 | 1200 | 1200 | 1 |
| 21 | 0.67 | 3 | 3000 | 300 | 10 |
| 22 | 0.67 | 15 | 2600 | 100 | 26 |
| 23 | 0.7 | 10 | 1500 | 300 | 5 |
| 24 | 0.7 | 20 | 2200 | 100 | 21 |
| 25 | 0.75 | 16 | 0 | 300 | 0 |
| 26 | 0.75 | 4 | 900 | 900 | 1 |
| 27 | 0.75 | 4 | 1800 | 600 | 3 |
| 28 | 0.75 | 4 | 3300 | 100 | 33 |
| 29 | 0.78 | 23 | 0 | 200 | 0 |
| 30 | 1 | 1 | 1800 | 1800 | 1 |
| 31 | 1 | 2 | 1800 | 900 | 2 |
| 32 | 1 | 1 | 2400 | 1200 | 2 |
| 33 | 1 | 1 | 3300 | 300 | 11 |
| 34 | 1 | 3 | 3000 | 200 | 15 |
| 35 | 1 | 1 | 3500 | 100 | 35 |

After the game, a post-interview is conducted for each subject by a research assistant (RA), who ask a specific list of questions in Japanese language to elicit the motivational factors affecting subject choices in ISDG (i.e., structured interview). The questions in the post-interview are:

- Please rank the following factors from the most influential factors (1) to the least influential factors (4) on your decision (rank order question)

  - History
  - $X$
  - $D$
  - $\frac{X}{D}$
- What did you care about or consider when you chose options *A* or *B*? (Open-ended question)
- Did the history affect your decision? How? (Open-ended question)
- Did the magnitude of gain *X* affect your decision? How? (Open-ended question)
- Did the difference *D* affect your decision? How? (Open-ended question)
- Did the ratio of $\frac{X}{D}$ affect your decision? How? (Open-ended question)
- What advice would you give if you could give advice to each participant in the experiment from the position of a person who will not have any gain from the game? (Open-ended question)

- How far was the advice in the previous question from your actual behaviors in the game? (Open-ended question)

Each post-interview is recorded and the records are transcribed and translated to English by the first and second authors. The questions of the post-interview are used to elicit the opinions and ideas of the subjects in ISDG. The interview method has been applied by previous research, which is known to be reliable. While, the theory of saturation can express the validity of the data because several subjects keep repeating the same concepts and sentences leading to the confidence that we collected the correct concept and ideas in both basic ISDG and FAB treatments [28,29].

## 2.2. Experimental Procedures

Figure 1 shows the procedures of one-person ISDG and the post-interview in one session for basic ISDG and FAB treatments. Upon arrival to a gathering room, each subject picks up a lottery that determines her experimental ID. Then, subjects are taken to two different designated rooms based on their experimental IDs. In basic ISDG treatment, each subject reads the experimental instructions and listens to an oral presentation made by an experimenter about basic one-person ISDG, where neutral terminology is used for the explanations. After that, each subject experiences 36 situations of basic one-person ISDG treatment in a shuffled order. Each subject makes her decision by choosing between options *A* and *B* in each of the situations. When a subject finishes the decisions in all 36 situations, he/she is informed of the situation number that corresponds to the binding situation to determine her final payoff of one-person ISDG. Then, each subject moves to a different room with an RA to be interviewed individually. Each RA is trained to ask the questions in the same order and in a neutral way to avoid the effect of the different RAs.

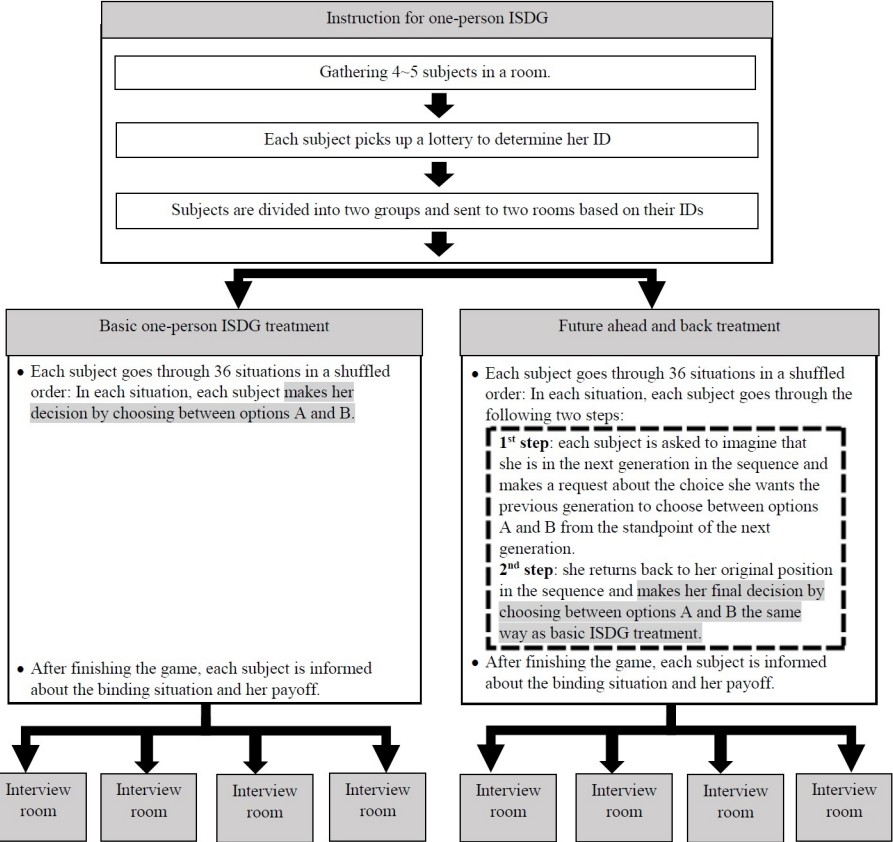

**Figure 1.** Procedures of one-person ISDG and the post-interview in one session.

In FAB treatment, subjects follow the same steps of basic ISDG in addition to the perspective taking step. In each situation, a subject is asked to imagine that he/she is in the position of the next generation in a sequence. From that position, he/she makes a request to the previous generation about the choice he/she wants the previous generation to choose. After that, he/she returns back to her original position in a sequence and makes her final decision through choosing between options *A* and *B*. The time for each session varies between basic ISDG and FAB treatments. One session in basic ISDG treatment consists of two parts and takes approximately 75 min. In the first part, subjects experience one-person ISDG for 40 min. After that, they are interviewed as the second part for 7~10 min. One session in FAB treatment also consists of two parts and takes approximately 90 min. In the first part, subjects experience one-person ISDG for 55 min that is longer than that of basic ISDG treatment due to additional procedures in FAB treatment. After that, each subject is interviewed individually by an RA for 7~10 min. Each subject participates in one session only and is paid 3000 JPY ($\approx$28 USD) on an average.

## 3. Results

Table 2 summarizes the statistics for the experimental results in basic ISDG and FAB treatments. The number of subjects who participates in the basic ISDG and FAB treatments is 56 and 48, respectively. The total number of choices for all subjects is 2012 and 1728 in the basic ISDG and FAB treatments, respectively. Approximately 33.7 % and 40.2 % out of the total number of choices are option *B* in the basic ISDG and FAB treatments, where the percentage of choosing option *A* is around 66.3 % and 59.8 %, respectively. This indicates that the majority of the choices are option *A* in both treatments, while option *B* is chosen more in FAB treatment than in basic ISDG treatment. To confirm the difference statistically, a chi-square test is run with the null hypothesis that the frequency distribution of the observations for choosing options *A* and *B* are the same between basic ISDG and FAB treatments and the null hypothesis is rejected at 1 % significance level ($\chi^2 = 16.75, P < 0.01$). Refer to Shahen et al. [30] for more information about the results of subjects' decisions in the experiment. This shows that there is a difference between the choices of options *A* and *B* in basic ISDG and FAB treatments, implying that there might be differences in the motivations between the treatments.

**Table 2.** Summary of statistics.

|  | Basic ISDG Treatment | FAB Treatment |
|---|---|---|
| Total no. of subjects | 56 | 48 |
| No. of situations per subject | 36 | 36 |
| Total number of choices | 2012 | 1728 |
| Choices of option *A* | 1333 (66.3 %) | 1033 (59.8 %) |
| Choices of option *B* | 679 (33.7 %) | 695 (40.2 %) |

The post-interview contents are analyzed to explore the motivational factors during the decision-making process in ISD. In the post-interview, the responses of the question "Please rank the following factors from the most influential factor (1) to the least influential factors (4) on your decision" are analyzed and projected in Figure 2. This figure presents a bar chart for the average self-reported ranking of the factors affecting subject's decisions in basic ISDG and FAB treatments. There is no significant difference in the average ranking of history between the basic ISDG and FAB treatments. On the other hand, the ranking of *X*, *D* and $\frac{X}{D}$ factors are different between the treatments. In FAB treatment, subjects consider *D* and *X* less ($\frac{X}{D}$ more) influential in their decisions in comparison to basic ISDG. To statistically confirm the difference, a Mann–Whitney test is run with the null hypothesis that the distribution for the ranking of each factor i.e., history, *X*, *D* and $\frac{X}{D}$ between basic ISD and FAB treatments are the same. The null hypothesis is rejected for *X* and $\frac{X}{D}$ at 5 % significance level and it can not be rejected for history and *D* factors, showing that subjects' decisions under FAB treatment are more influenced by $\frac{X}{D}$ in comparison to basic ISDG. This indicates that FAB treatment induces subjects

to consider future generations because $\frac{X}{D}$ represents the number of future generations who can have a positive payoff before resource extinction.

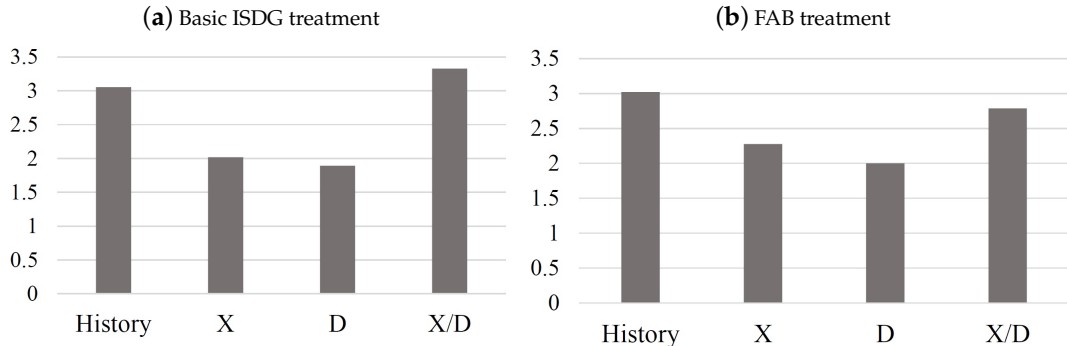

**Figure 2.** The average self-reported ranking of factors affecting subject's decisions in ISDG.

The contents of the remaining questions in the post-interview are analyzed using KH Coder 3.0 [31]. This program is a free text-mining software that provides ways for quantitative analysis of texts to quantify the relationship between ideas and words in the post-interview. This software uses a process called tokenization to demarcate and classify the words of each sentence and arrange each word as a unite of analysis (i.e., token). The number of tokens in basic ISDG and FAB treatments is 15,122 and 11,483, respectively. All the tokens are used in the analysis, while the nouns and proper nouns are considered as the keyword in the analysis. Table 3 presents the number of keywords in basic ISDG and FAB treatments, which are 2491 and 1892, respectively. The frequencies of the top ten keywords in the basic ISDG and FAB treatments are presented in Table 3.

**Table 3.** The frequency of the top ten keywords in the post-interview.

| Basic ISDG Treatment | | FAB Treatment | |
|---|---|---|---|
| **Keywords** | **N** | **Keywords** | **N** |
| option_B | 273 | option_B | 201 |
| option_A | 247 | option_A | 176 |
| difference_D | 127 | next_person | 98 |
| next_person | 105 | gain | 93 |
| gain | 102 | difference_D | 87 |
| previous_person | 94 | gain_X | 64 |
| choice | 84 | option | 57 |
| point | 79 | X_over_D ($\frac{X}{D}$) | 47 |
| gain_X | 76 | person | 47 |
| decision | 73 | ratio | 47 |
| Total # of keywords | 2491 | | 1892 |

Subjects in the basic ISDG and FAB treatments tend to mention the keywords "option B" and "option A" more frequently than other keywords. In basic ISDG treatment, the most frequently mentioned keywords are "difference *D*", "next person", "gain", "previous person", "choice", "point", "gain *X*" and "decision". This result indicates that subjects consider some factors such as the difference *D*, the next person, the gain and the previous generation important in their decision-making process. On the other hand, in FAB treatment, the most frequent keywords are "next person", "gain", "difference *D*", "gain *X*", "option", "$\frac{X}{D}$", "person" and "ratio". This finding means that subjects in FAB treatment consider the next person, the gain, the difference *D* and $\frac{X}{D}$ influential factors for their decisions. The frequency of these keywords shows that some terms appear in both treatments for example "difference *D*", "next person" and "gain", while other terms appear only in one treatment for instance "previous person" in basic ISDG treatment and "$\frac{X}{D}$" in FAB treatment. These results imply

that there can be differences in the motivational factors that affect the subject decisions in the basic ISDG and FAB treatments.

A co-occurrence network is presented in Figures 3 to 6 to analyze how subjects mention the keywords in the post-interview. The networks show the frequency of each term (i.e., keyword) used in the subjects' answers and the relationship between them. Each term is located in a circle, which is called a node. The frequency of terms represented by the size of the node in the co-occurrence network, while the color of nodes represents the group of terms that appears together in the same sentence or cluster. The relationship between terms is represented by a connecting line between the nodes, which is called an edge. The number on each edge represents Jaccard coefficient, which is a measurement of the co-occurrence of terms. Jaccard coefficient is calculated as follows $J(X, Y) = \frac{(X \cap Y)}{(X \cup Y)}$, where $X$ and $Y$ represents two different terms [32]. When there are no relationships between two terms, then $J(X, Y)$ equals zero and there is no edge connecting these terms. In the co-occurrence networks, the graphs show only the edges, which have a coefficient more than 0.94 % because showing all the edges will make the graph hard to interpret. When the relationship between the two terms is weak, the edge appears as a dotted line. The weak relationship between two terms means that these terms do not appear directly together, but they appear with some other terms between them. Four co-occurrence networks are presented by the content analysis of the post-interview. The first two networks represent the whole post-interview contents to show the relationship and the connection between the terms in subject responses, while the last two networks present only the keywords.

Figure 3 shows the co-occurrence network for the post-interview in basic ISDG treatment. The co-occurrence network shows that the node of the term "I" is the biggest in size with a frequency of 1000 and connected indirectly to a long chain of nodes, which allow us to understand the context. This node has edges with four nodes i.e., "not", "think", "choose" and "be", where the edges are strong with "be" and "choose" nodes with coefficients of 62 % and 50 %, respectively. These two nodes co-occur with a chain of recurrent nodes, indicating that several subjects repeat such a sequence of words. "Choose" node shares edges with three nodes "option *A*", "option *B*" and "when" with coefficients of 55 %, 54 % and 31 %, respectively. Out of these three nodes, "when" node has edges with two nodes "negative" and "large" with a coefficient of 23 %. Consequently, "large" node has an edge with "difference D" node with a coefficient of 24 %. On the other hand, "be" node is indirectly connected to a chain of six nodes "my", "decision", "make", "choice", "previous person" and "select" with coefficients of 26 %, 25 %, 36 %, 31 %, 23 % and 24 %, respectively. Other nodes in this co-occurrence network do not provide meaningful information because the size of nodes is small and there is no long chain of nodes to understand the context. This co-occurrence network shows that some sentences are frequently repeated in the post-interview such as "I am making my decision based on the choice of the previous person" and "I choose option *A* or *B* when the difference *D* is large", indicating that subjects in basic ISDG treatment tend to think about the difference *D* and the previous generations' choices when they are making their decisions.

Figure 4 shows the co-occurrence network for the post-interview in FAB treatment. The node of the term "I" is the biggest in size with a frequency of 800 and is connected with a long chain of nodes. This node has strong edges with four nodes i.e., "not", "think", "choose" and "be" with coefficients of 41 %, 48 %, 48 % and 60 %, respectively. The relationship is explained between the nodes connected to a long chain such as "choose" node. This node shares edges with "option *A*" and "option *B*" with coefficients of 55 % and 54 %, respectively. Consequently, "option *B*" node has edges with one node "when" with a coefficient of 33 %. "When" node shares edges with two nodes "difference D" and "negative" with a coefficient of 24 %. "Difference D" has edges with three nodes "large", "next person" and "small" with coefficients of 22 %, 24 % and 25 %. This long chain of nodes represents the thoughts that subjects consider while choosing option *B* in FAB treatment. This co-occurrence network shows that some sentences are frequently repeated in the post-interview for example "I choose option *B* when the difference *D* is large or small for the next person", suggesting that subjects are considering the difference *D* and the next person when they choose option *B*.

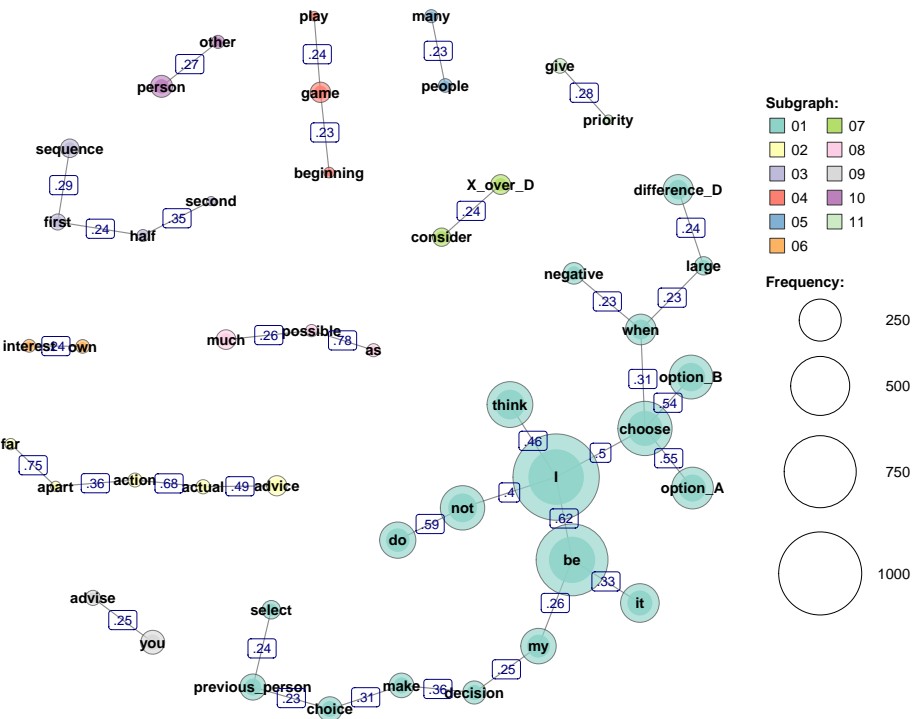

**Figure 3.** Co-occurrence network for the post-interview in basic ISDG treatment.

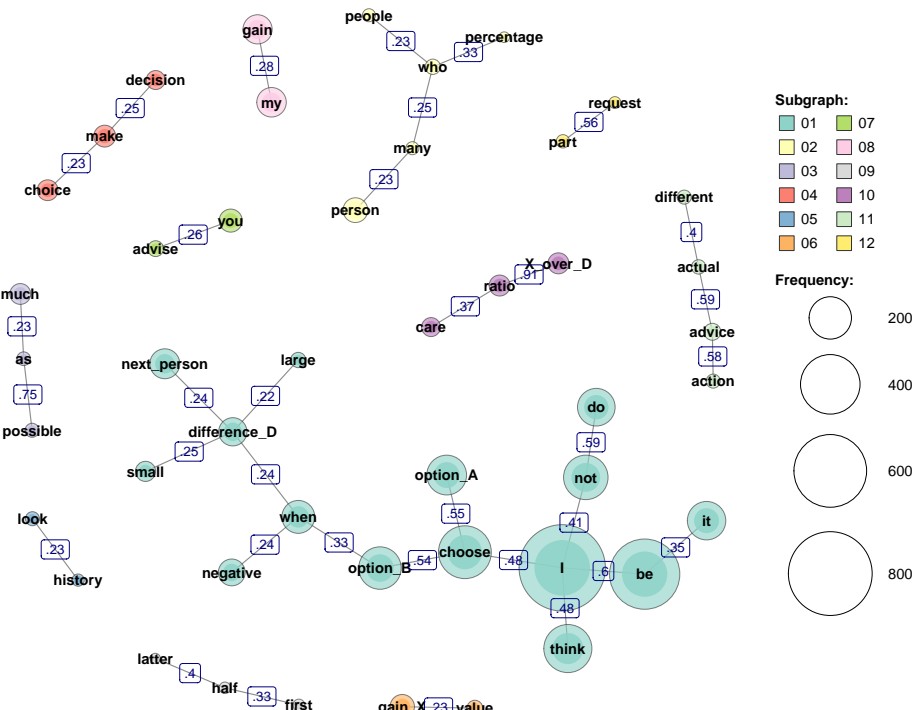

**Figure 4.** Co-occurrence network for the post-interview of subjects in FAB treatment.

Figures 5 and 6 are the co-occurrence networks of the keywords in the post-interview, which present the relationship between the factors that affect subject decisions in basic ISDG and FAB treatments, respectively. Figure 5 shows that the nodes of options *A* and *B* are the biggest with the highest number of edges, indicating that subjects try to illustrate their motivations to choose one

option. The node of "option *A*" has edges with three nodes "next person", "difference *D*" and "point" with coefficients of 18 %, 21 % and 15 %, respectively. This indicates that subjects choose option *A* because they think about the difference *D*, the points they might get and/or the next person's gain. "Difference *D*" node is connected with a long chain of nodes, but edges of this chain are weak. On the other hand, the node of "option *B*" has edges with three nodes "previous person", "difference" and "time" with coefficients of 18 %, 18 % and 10 %. The node of "previous person" is connected with a chain of nodes, indicating that subjects can be motivated to select option *B* by considering previous generations' choices. This indicates that the motivational factors for subjects to consider options *A* and *B* is mostly the difference *D* and the previous person's choices, respectively.

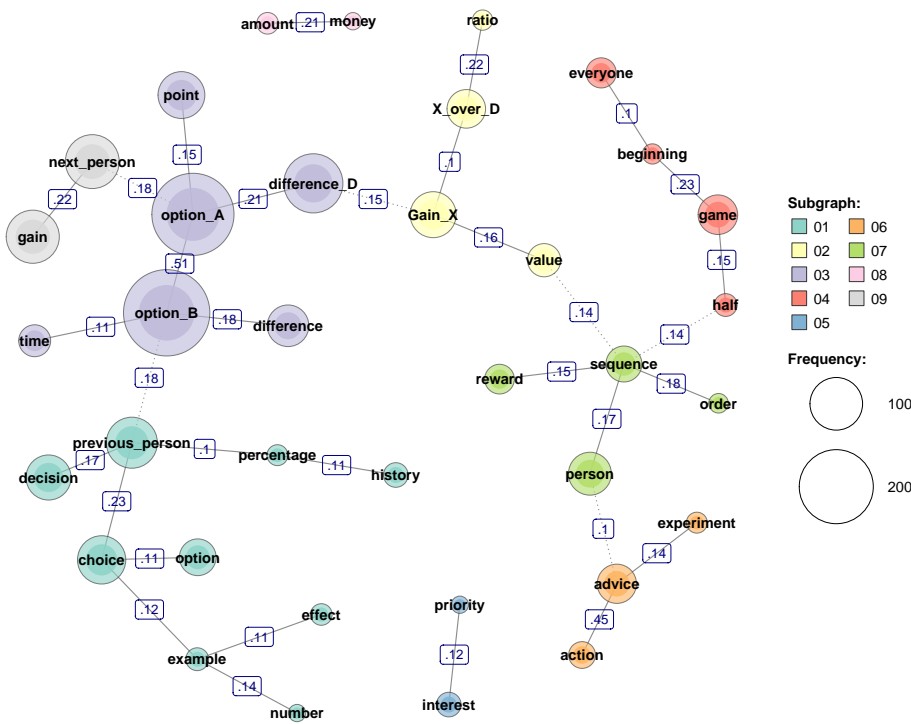

**Figure 5.** Co-occurrence network for the keywords in the post-interview of subjects in basic ISDG treatment.

In Figure 6, the nodes of "option *A*" and "option *B*" are the biggest nodes with the frequency of 200, indicating that subjects try to illustrate their motivations to choose one option. The node of "option *A*" has edges with two nodes "time" and "difference" with coefficients of 14 % and 13 %, respectively. This indicates that subjects choose option *A* when they think about the time limitations and/or the difference *D* and because the chains are not long it is difficult to understand the context of these words. On the other hand, the node of "option *B*" has edges with two nodes "next person" and "person" with coefficients of 21 % and 18 %. The node of "next person" is connected with three nodes (i.e., "choice", "gain" and "difference") and these nodes have edges with others. This indicates that there is a diversity in the motivational factor to select option *B* in FAB treatment and these factors are usually related to the next person. "Option *B*" node has also edges with node "person" with a coefficient of 18 %, but this does not provide a meaningful indicator as the "person" term does not include any keywords. These results indicate that when subjects think about choosing option *B* in FAB treatment, they consider the possible consequence of their decisions on the next generation.

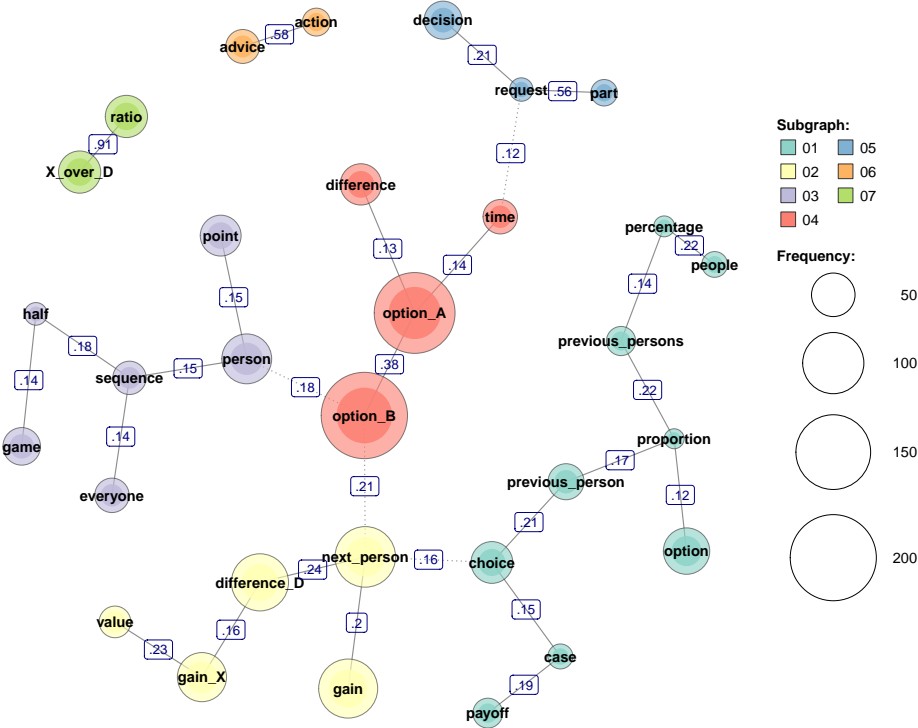

**Figure 6.** Co-occurrence network for the keywords in the post-interview of subjects in FAB treatment.

The findings indicate that FAB might be effective in changing individual motivations as well as behaviors to consider IS, which can be considered in line with previous literature. Some scholars implemented future design mechanisms in the field by taking the perspective of future generations to find solutions for intergenerational issues such as financial sustainability, forestry management and water supply management in Japan [33–35]. Nakagawa et al. [33] implement a case study experiment using the general public as a subject pool to study the effect of intergenerational retrospective viewpoint as a future design mechanism on the process of policy formulation for forest management in Kochi city in Japan. Shahrier et al. [20] apply FAB mechanism in urban and rural areas in Bangladeshi to study the behaviors towards IS, demonstrating that FAB is effective in maintaining IS.

Overall, the findings show that individuals have different motivations and ideas to consider one choice under the basic intergenerational sustainability dilemma game (ISDG) and future ahead and back (FAB) treatments. In basic ISDG treatment, individuals are usually past-oriented because they think about the effect of previous generations' decisions. On the other hand, FAB treatment induces individuals to be future-oriented because individuals become motivated to choose the sustainable option by considering the possible consequences of their actions on future generations. These results suggest that individuals focus on the past in the status quo, leading to repeating the same choice of history by being sustainable or unsustainable. However, one way to bring new motivational factors to consider IS can be the introduction of some mechanisms to take the perspective of future generations such as FAB [1,2,14].

## 4. Discussion

The results indicate that the history of the previous generation's choices mostly influences current generation decisions in basic ISDG. Literature in anthropology and social psychology have shown that social-learning can lead individuals to copy the decision of others through observation [36–38]. Thus, social-learning is conjectured to lead individuals in the current generation to focus on previous generations' choices to imitate it in basic ISDG treatment. The results also indicate that when the

current generation takes the perspective of future generations in FAB treatment, he/she considers the possible consequence of his/her decision.

Cooper [39] claims that "cognitive dissonance" emerges from the experience of two or more different conflicting psychological representations in a decision-making situation, which affects individual decisions. In this sense, "cognitive dissonance" is argued to be a possible reason because under FAB treatment each individual experiences two representations of the current and future generations where there is a conflict of interests among generations. Another possible reason for this result could be "empathy" towards future generations because it may influence altruistic motivations towards unknown others [40–42]. In FAB treatment, "empathy" is triggered by taking the position of future generations, leading the current generation to consider the possible consequence of their decisions on the intergenerational sustainability (IS).

This paper suggests that understanding the motivational factors enables us to understand how humans think and make decisions to design a social mechanism that can help individuals to change their behaviors and preferences. In our finding, individuals are past-oriented in basic ISDG and without changing people's attitudes and behaviors sustainability is threatened. Applying FAB mechanism and taking the perspective of future generations lead to changes in motivations that induce individuals to maintain IS. Thus, understanding the motivations of individuals is essential to design new social mechanisms and implement some practices in a society. In several occasions, the policy are formulated mostly in developing countries to solve some problems without considering distant future consequences, which leads to failure of policy impact to contemplate such long term repercussions. For example, Sida aids for electricity and natural resources in India and Zambia that fail to have a sustainable impact in the local communities due to lack of understanding of the motivations for investment [43]. Therefore, this study proposes that the motivational factor should be examined to design and implement a proper mechanism to address such short and long term problems.

To design a sustainable society, it is evident that people's motivations, values and beliefs should be analyzed besides the way individuals interpret sustainability problems. This requires quantitative and qualitative analysis to understand the risks and the opportunities for designing such a sustainable society [44]. In developed countries, the infrastructures of energy, water, transportation and urban planning are well developed. However, they were built in the past when sustainability was not threatened. Thus, the perspective of future generations has not been considered at that time. Nowadays, many developed countries start to develop societies to be sustainable, however in several instances taking prospective of future generations has been broadly missing. Recently some prefectures in Japan are developing sustainable cities and managing forestry and water supply through practicing future design by conducting deliberative workshops to consider the motivations of individuals [5,33–35]. Overall, our study provides a new avenue for policy development in both developing and developed nations by understanding the motivations and taking the perspective of future generations.

## 5. Conclusions

This paper addresses the motivational factors of individual decisions in the "intergenerational sustainability dilemma" (i.e., ISD) by hypothesizing that these factors are different with or without perspective-taking of future generations. Thus, a one-person ISD game (ISDG) along with a post-interview is instituted. In addition, a future ahead and back (FAB) mechanism is applied as a treatment for perspective-taking of future generations. By analyzing the post-interview contents with text-mining techniques, the paper find that individuals mostly consider how previous generations had behaved in basic ISDG as the main motivational factor. However, individuals in FAB treatment are induced to put more weights on the possible consequences of their decisions for future generations as motivational factors. The findings indicate that perspective-taking of future generations through FAB mechanism enables people to change not only their decisions but also motivational factors, enhancing ISD.

Some limitations are noted along with future avenues of research. This study uses a structured interview to elicit the ideas and motivational factors during the decision making process. This interview might not give the chance for the individuals to fully express other moral, cultural and environmental factors that could have some influence on their decisions in the experiment. Thus, future research can conduct in-depth unstructured interviews, which enables us to understand how the cultural and moral factors affect the behaviors in the economic experiment. These caveats notwithstanding, it is believed that this paper is an important first step in experimental economic research that addresses the motivations in intergenerational sustainability. This approach of quantitative analysis of qualitative data could provide a new way for experimental research to statistically analyze ideas and concepts and motivational factors for human behaviors in experiments.

**Author Contributions:** Conceptualization, M.E.S., and K.K.; data curation, M.E.S., W.M. and K.K.; formal analysis, M.E.S., W.M. and K.K.; funding acquisition, K.K. and T.S.; investigation, M.E.S., W.M. and K.K.; methodology, M.E.S. and K.K.; project administration, M.E.S. and K.K.; resources, M.E.S. and K.K.; supervision, M.E.S. and K.K.; validation, K.K.; visualization, M.E.S., W.M. and K.K.; writing—original draft, M.E.S. and K.K.; writing—review and editing, M.E.S., T.S. and K.K. All authors have read and agreed to the published version of the manuscript.

**Funding:** We are grateful to the financial supports from the Japan Society for the Promotion of Science (JSPS) as the Grant-in-Aid for Scientific Research A (17H00980) and Kochi University of Technology.

**Acknowledgments:** The authors thank anonymous referees, Makoto Kakinaka, Hiroaki Miyamoto, Yoshinori Nakagawa, Raja Rajendra Timilsina, Khatun Mst Asma, Pankaj Koirala and Jingchao Zhang for their helpful comments, advice and supports.

**Conflicts of Interest:** The authors declare no conflict of interest.

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
