# Peer review of "Motivational Factors in Intergenerational Sustainability Dilemma: A Post-Interview Analysis"

_sustainability, doi:10.3390/su12177078_

Round 1

Reviewer 1 Report

The paper experimentally addresses a very interesting aspect related to intergenerational sustainability dilemma, namely the perspective taking of future generations where individuals of one generation are asked to take the position of the next generation and make a request about the choice that they would the current generation to choose accordingly. They show that the future ahead and back (FAB) treatment indeed works by inducing treated individuals to put more weight on the possible consequences of their decisions for future generations as motivational factors.

The research topic is of extreme importance and the experiment is designed very carefully. Although the paper is well-written and I personally enjoyed reading it, I believe the paper will benefit from a revision that goes in the direction of stressing the value added of the analyses, and more importantly the policy relevance of the findings. The two points I’d like to see improvement on are:

- The introduction puts a lot of emphasis on the contribution of the paper above and over the existing literature which I believe is important, but I am afraid is not sufficient. The introduction as well as the conclusion need deeper discussion on the policy implications of putting the perspective of future generations into consideration when planning for the future and dealing with the ISD.

- The researchers also need to discuss which possible policy actions/tools (if any) could be useful in achieving this future perspective in real-life situations? To what extent these tools could work in different countries with different institutional structures and cultures. For example, would the future perspective approach (using FAB techniques) work the same way in developing vs advanced countries or for collectivistic vs individualistic societies. This needs to be addressed and discussed in more detail.

Author Response

Please find our reply in the attached file.

Reviewer 2 Report

This research is interesting. The intergenerational sustainability dilemma is used to discuss Motivational factors. This article is an inference method. It is used to predict whether there will be an impact between different generations under certain conditions. The inference method in this article is similar to Previously published articles. In general, the manuscript has been prepared very honestly.
The overall idea of ​​the paper is very interesting as well as the described research subject. However, the authors did not address several aspects suitably, thus the value of the paper decreased significantly.

Despite the proposal is very interesting, some issues need to be supplemented.
Please arrange the article according to the format of the journal.
The Introduction section should mention the present work. At the end of the introduction, it should mention the following sections in this paper briefly.
The number of experimental samples, representativeness, analysis process, and data results are difficult to obtain support for results. Based on the process of classifying to describe how was the data measured? Do the values ​​of reliability, validity, and verification worth to refer? So these also need to be described by authors.

Author Response

(The authors gave the same response as above.)

Reviewer 3 Report

This paper might have potential to make a useful contribution, but it is not ready for the publication as the research article or full-length article. The paper is mostly (only first part of introduction is correct) description of research which was conducted at University.

The paper prepared was not followed any single journal format. 

In this paper authors are focused on the research study and they upgraded existing methodology. Therefore, the methodology is the strongest part of the paper. However, process of conducted research is too long and confusedly explained. Results are presented clearly and analysed appropriately but in the same time conclusion is to short and does not exist clear implication of this research. Also, the limitations of this research are not given.

The paper should be written in the 3rd person.

Author Response

(The authors gave the same response as above.)

Round 2

Reviewer 1 Report

Thank you for taking care of the comments. The paper reads nice and I believe it provides useful insights to the topic. 

Author Response

Thank you very much for the comments.

Reviewer 2 Report

The paper may be provided useful insights to the readers in the journal of sustainability.

Author Response

Thank you very much for the comments.

Reviewer 3 Report

In the introduction it is mentioned, from line 22 until line 81, former researches but their implication is still not clear and from it is not clear connection whit this paper. This part needed deeper discussion which is going put this research in to clear perspective.

In the chapter 2 – “Methods and materials”; from line 76 until line 162 already are explained some results but just from line 164 until 169 are given basic data about researched sample. This order is not logical and confuses the reader.

In this part, from page 5, line 125 are given sentences with “ As if she is in the next generation, she is asked to..”, and it continues later. From this expression, is not clear who “she” is. Are the only ones females in this research or?

Thus, it should be overwritten.

Results are presented clearly but discussion is missing. Some of discussion parts are given in the conclusion but should be much more widely explained is separate section.  For example, from this part is not clear, why are now are economist are involved (page 14, line 338), they have not been mentioned earlier. This connection/implication is missing from the introduction.

The paper should be upgraded by discussion, for example: how this research could be implemented in the future real-live, and could this methodology be used in different countries and could it be the same for the developed and undeveloped countries?

The paper should be written in the 3rd person. – page 2, line 84 – “However, to our knowledge,..”

- page 2, line 82 – “Past literature has..”

Author Response

Please find the reply in the attachment. 
